# Single-Shot Vaccines against Bovine Respiratory Syncytial Virus (BRSV): Comparative Evaluation of Long-Term Protection after Immunization in the Presence of BRSV-Specific Maternal Antibodies

**DOI:** 10.3390/vaccines9030236

**Published:** 2021-03-09

**Authors:** Jean François Valarcher, Sara Hägglund, Katarina Näslund, Luc Jouneau, Ester Malmström, Olivier Boulesteix, Anne Pinard, Dany Leguéré, Alain Deslis, David Gauthier, Catherine Dubuquoy, Vincent Pietralunga, Aude Rémot, Alexander Falk, Ganna Shevchenko, Sara Bergström Lind, Claudia Von Brömssen, Karin Vargmar, Baoshan Zhang, Peter D. Kwong, María Jose Rodriguez, Marga Garcia Duran, Isabelle Schwartz-Cornil, Geraldine Taylor, Sabine Riffault

**Affiliations:** 1Department of Clinical Sciences, Swedish University of Agricultural Sciences, 750 07 Uppsala, Sweden; sara.hagglund@slu.se (S.H.); katarina.naslund@slu.se (K.N.); estermalmstrom@gmail.com (E.M.); 2Université Paris-Saclay, INRAE, UVSQ, VIM, 78350 Jouy-en-Josas, France; luc.jouneau@inrae.fr (L.J.); catherine.dubuquoy@gmail.com (C.D.); vincent.pietralunga@gmail.com (V.P.); isabelle.schwartz@inrae.fr (I.S.-C.); sabine.riffault@inrae.fr (S.R.); 3INRAE, PFIE, 37380 Nouzilly, France; olivier.boulesteix@inrae.fr (O.B.); anne.pinard@inrae.fr (A.P.); dany.leguere@inrae.fr (D.L.); alain.deslis@inrae.fr (A.D.); david.gauthier@inrae.fr (D.G.); 4INRAE, Université de Tours, ISP, 37380 Nouzilly, France; aude.remot@inrae.fr; 5Department of Chemistry-BMC, Uppsala University, 752 37 Uppsala, Sweden; alexander.falk@kemi.uu.se (A.F.); ganna.shevchenko@kemi.uu.se (G.S.); sara.lind@uadm.uu.se (S.B.L.); 6Department of Energy and Technology, Swedish University of Agricultural Sciences, 750 07 Uppsala, Sweden; claudia.von.Bromssen@slu.se; 7Department of Biomedicine and Veterinary Public Health, Swedish University of Agricultural Sciences, 750 07 Uppsala, Sweden; karin.vargmar@slu.se; 8Vaccine Research Center, National Institute of Allergy and Infectious Diseases, National Institutes of Health, Bethesda, MD 20892, USA; baoshan.zhang@nih.gov (B.Z.); pdkwong@nih.gov (P.D.K.); 9Inmunología y Genética Aplicada, S.A. (INGENASA), 28037 Madrid, Spain; mjrodriguez@ingenasa.com (M.J.R.); mgarcia@ingenasa.com (M.G.D.); 10The Pirbright Institute, Woking, Surrey GU24 0NF, UK; geraldine.taylor@pirbright.ac.uk

**Keywords:** respiratory syncytial virus, bovine, neonatal, subunit vaccine, pre-fusion, live-attenuated vaccine, deleted SHrBRSV, one-shot, duration of protection, correlate of protection

## Abstract

The induction of long-lasting clinical and virological protection is needed for a successful vaccination program against the bovine respiratory syncytial virus (BRSV). In this study, calves with BRSV-specific maternally derived antibodies were vaccinated once, either with (i) a BRSV pre-fusion protein (PreF) and Montanide^TM^ ISA61 VG (ISA61, *n* = 6), (ii) BRSV lacking the SH gene (ΔSHrBRSV, *n* = 6), (iii) a commercial vaccine (CV, *n* = 6), or were injected with ISA61 alone (*n* = 6). All calves were challenged with BRSV 92 days later and were euthanized 13 days post-infection. Based on clinical, pathological, and proteomic data, all vaccines appeared safe. Compared to the controls, PreF induced the most significant clinical and virological protection post-challenge, followed by ΔSHrBRSV and CV, whereas the protection of PreF-vaccinated calves was correlated with BRSV-specific serum immunoglobulin (Ig)G antibody responses 84 days post-vaccination, and the IgG antibody titers of ΔSHrBRSV- and CV-vaccinated calves did not differ from the controls on this day. Nevertheless, strong anamnestic BRSV- and PreF-specific IgG responses occurred in calves vaccinated with either of the vaccines, following a BRSV challenge. In conclusion, PreF and ΔSHrBRSV are two efficient one-shot candidate vaccines. By inducing a protection for at least three months, they could potentially improve the control of BRSV in calves.

## 1. Introduction

Bovine respiratory syncytial virus (BRSV) causes respiratory disease in cattle, with important direct and indirect animal health and economic consequences [1]. BRSV is an enveloped, nonsegmented, negative-stranded RNA virus that belongs to the *Orthopneumovirus* genus within the *Pneumoviridae* family [2]. This virus spreads easily and causes high morbidity and severe disease, which impairs cattle welfare and production [3,4,5].

The prevention of BRSV and other respiratory infections in cattle is based on biosecurity and husbandry management, which may include vaccination. The type of production, rearing system, and activities of cattle professionals linked to the cattle density in a geographic region all impact on the possibility of implementing appropriate biosecurity measures. Consequently, such factors influence the need for vaccination to prevent respiratory disease. Commercial vaccines are available, but their efficacy is not always satisfactory. The duration of protection of available vaccines may be rather short, especially when immunization is performed in the presence of maternally derived antibodies (MDA) that impede the calf’s response to vaccination [6,7,8]. Other reasons that could contribute to this lack of efficacy may be related to the vaccine itself and its storage conditions or to the vaccination procedure. Due to the necessity to restrain the heads of animals and the risk of animals sneezing, it can be problematic to administer a standardized vaccine dose by the intranasal (i.n.) route. Furthermore, vaccination protocols are not always followed when two doses or more are required to induce protection. The use of a one-shot parenterally administered vaccine reduces the vaccination costs and increases the feasibility of the immunization schedule. An efficient vaccine in calves with BRSV-specific MDA would be of great interest, especially if the induced protection is of a long duration.

Human RSV (HRSV) is also a major respiratory pathogen, both in young children and the elderly, but a commercial vaccine is not yet available. In the 1960s, a formalin-inactivated HRSV vaccine failed to protect young infants against infection and primed for enhanced respiratory disease following RSV infection [9]. This raised safety concerns that have severely delayed vaccine development. Despite decades of extensive research, the development and evaluation of a new HRSV vaccine is still ongoing. As a result of similar epidemiological, clinical, and immunopathological characteristics, BRSV in calves represents a very pertinent model of HRSV, with the potential to complement small animal models in research [10]. Both the safety and efficacy aspects of RSV vaccines can be studied, including the duration of protection. While safety issues are still central for HRSV vaccine research, knowledge on the duration of protection is essential for the design of vaccination programs. The present study compares two of the vaccine strategies currently considered the most promising for infant vaccination, namely one parenteral subunit vaccine and one mucosal live-attenuated vaccine designed by reverse genetics.

The calf model has been used to study a recombinant pre-fusion F protein of BRSV (PreF), produced by transfected human cells, as a candidate subunit vaccine. Previous studies demonstrated that 50 µg of PreF in a water-in-oil adjuvant, ISA71 VG (Seppic, Paris, France), administrated intramuscularly (i.m.) twice at a four-week intervals to calves with no or little BRSV-specific MDA, induced strong virological protection against the BRSV challenge, which was superior to that induced by the same formulation of a post-fusion (PostF) form of the F protein [11]. The superiority of PreF compared to PostF was attributed to the strong induction of neutralizing antibodies. In a subsequent experiment with a more stringent situation of calves with BRSV-specific MDA, a single i.m. injection with 100-µg PreF in ISA61 VG adjuvant (Seppic, Paris, France), alone or combined with HRSV N protein–nano rings (N Nano rings), induced near complete clinical and virological protection against the challenge with virulent BRSV four weeks later [12]. Protection was again found to be associated with a strong induction of neutralizing antibodies. Based on the cost of production and the fact that the level of protection was not significantly different with or without N nano rings, we further aimed to evaluate PreF in ISA61 for protection, three months after a single i.m. injection in calves with BRSV-specific MDA. This time, PreF was compared with an experimental i.n. vaccine composed of live-attenuated BRSV lacking the SH gene, ΔSHrBRSV [13,14], and with an i.n. live-attenuated commercial vaccine (CV). Calves were challenged, as described previously [15], with the Snook strain of BRSV, three months after vaccination, since this is a time interval after which a lack of vaccine-induced protection was reported [8] and is commonly observed in the field (Valarcher, unpublished observations). Finally, another objective was to identify correlates of protection that may be predictive of protection three months after vaccination. This would be of interest to reduce the number of experimental animals used to evaluated BRSV vaccines and to be able to tailor vaccinations to targeted groups of animals in the field. Animals that are not adequately protected following a natural infection or vaccination could be identified and reimmunized. Such a tailored use of a vaccine that induces virological protection with a long duration could be implemented to decrease virus circulation between herds to a level at which sanitary measures will be sufficient to control the disease.

In this paper, we report the results to predict and evaluate the safety and efficacy of (i) PreF combined with ISA61 administered i.m. (PreF), (ii) ΔSHrBRSV administered i.n., (iii) a commercial live-attenuated vaccine administered i.n. (CV), and (iv) a placebo consisting of ISA61 alone administrated i.m. (control). A single vaccination was performed in calves with BRSV-specific MDA, three months prior to the BRSV challenge.

## 2. Materials and Methods

### 2.1. Calves and Vaccination Protocol

Twenty-four male calves of the Prim Holstein (*n* = 17) or Normand breeds (*n* = 7) were born during a period of one month in a conventional herd at the INRAE du Pin facilities (Gouffern En Auge, France) and moved to biocontainment facilities at INRAE Val de Loire (PFIE, Nouzilly, France) at the age of 5–16 days. One week before vaccination, they were allocated to four groups of six calves, based on their breed; age (mean age 29 days and range 17–49 days); and the level of BRSV-specific serum MDA on arrival at the PFIE facilities (mean 75%, 75%, 66%, and 70% of the corrected optical density (COD) of positive control sera, for controls or PreF-, ΔSHrBRSV, and CV-immunized calves, respectively). At vaccination (when the mean age of the calves was 36 days, range 24–56 days), one group of calves was immunized with 100-μg recombinant pre-fusion “DS2-v1” BRSV F protein, prepared as described previously [11], and adjuvanted with Montanide^TM^ ISA 61 VG (ISA61, kindly provided together with the IKA Ultra Turrax Tube Drive Mixer by SEPPIC S.A., Puteaux, France) in a 60/40 (*v*/*v*) proportion of adjuvant/protein in a 2-mL volume per dose. Another group received a placebo injection i.m. of ISA61 and PBS (60/40 *v*/*v*) in 2-mL volume per dose. Two groups of calves were vaccinated intranasally (i.n.) with either 5 × 10^6^ plaque forming units (PFU) of ΔSHrBRSV (2.5 mL in each nostril) or a commercial vaccine (CV) containing BRSV strain 375 (one dose 2 mL, 10^5.0^–10^7.2^ 50% tissue culture infective dose (TCID_50_), Rispoval intranasal, Batch 244787, Zoetis, Malakoff, France). The intranasal immunizations were performed using a mucosal atomization device (The MAD^®^, MedTree, Telford, UK). Groups were housed in different buildings to avoid cross-immunization following i.n. vaccination with the live-attenuated vaccine strains. The PreF-immunized calves and the controls were housed in the same building but in separate pens. Calves were monitored twice-daily after vaccination (temperature, appetite, and local reaction at the site of i.m. injection). Nasal swabs were collected every two days for 10 days to monitor live-attenuated virus replication. Three weeks after vaccination, all calves were moved to the same building with positive pressure, each group allocated to a separate pen. One week before challenge, calves were moved to level 2 bioconfined facilities.

### 2.2. Experimental Design, Challenge, Clinical Records, and Post-Mortem Analyses

The overall experimental design is presented in Figure 1. Briefly, calves were immunized on day (D)0 post-vaccination (pv) and were challenged with BRSV on D92 pv/D0 post-infection (pi). Calves were monitored until D13 pi when the experiment was terminated and necropsies were performed. Based on the high level of clinical signs of disease induced by the Snook strain of BRSV in previous studies, this virus strain was used to test the vaccines [15]. The challenge consisted of 1 × 10^4^ PFU of the virus, delivered by nebulization as previously described [15]. Clinical signs were monitored twice daily from D-2 pi until D13 pi, and were scored as described in Blodörn et al. (2015) [15], with slight modifications. Thirteen days post-challenge, all calves were euthanized by overdose of pentobarbital (140 mg/kg intravenously) (Doléthal, Vetoquinol, Lure, France), followed by exsanguination. The lungs were removed from the thoracic cage for observations and quantification of the macroscopic lesions by observation and palpation and for sample collection. The extent of lung lesions were drawn on a lung chart and the percentage of lesions was calculated using Image J [16] (free software, version 1.52a).

### 2.3. Sample Collection, Preparation and Storage

Blood, nasal swabs, nasal tampons, and bronchoalveolar lavage (BAL) were collected as shown in Figure 1 and were prepared and stored as described previously [17,18]. For PBMC isolation, citrate was used as anticoagulant. BAL was performed on calves under local anesthesia (Lidocain 20 mg/mL spray and gel, Astra Zeneca, Stockholm, Sweden), using sterile silicone tubing, D-2/D-1 pi and D6/D7 pi. Briefly, a 40–50-cm-long tube of 12 mm in diameter was inserted in one nostril and placed in the upper part of the trachea. Thereafter, a 1–1.5-m-long tube of 6 mm in diameter was pushed through the first tube until a bronchus was blocked, when 50 mL of NaCl was injected and immediately aspirated using a syringe attached to the tubing. This was repeated once and thereafter, any remaining BAL liquid was collected with a third syringe. The BAL was pooled and filtered through two layers of a sterile gauze, and BAL cells and supernatant were separated by centrifugation, as previously described [17,19]. The supernatants were stored at −75 °C, prior to mass spectrometric analyses. At necropsy on D13 pi, after quantification of the extent of the lung lesions, BAL and lung tissue were collected, prepared and stored as described previously [14,17,20], prior to virus isolation from BAL cells, BRSV-RNA quantification in BAL cells, and mass spectrometric analyses of BAL supernatants. Tissue samples were collected from areas of pneumonic consolidation if present, from each of the three lobes of the right lung and the accessory lobe and were fixed in 10% neutral buffered formalin (approx. 4% formaldehyde) (Sigma-Aldrich, St. Quentin Fallavier, France).

### 2.4. Detection of BRSV RNA and Live BRSV 

BRSV RNA coding for the F protein in nasal secretions and BAL cells corresponding to 10 mL of BAL, was quantified by Taqman RT-PCR as previously described [21]. The unit TCID_50_ equivalent (TCID_50_ eq.) was used since the standard curve in this assay was based on a BRSV infected cell lysate with a known titer (10^5.8^ TCID_50_). Virus titers were determined by an infectious focus assay on Madin-Darby Bovine Kidney (MDBK) cells. Samples were diluted 10-fold in Gibco MEM + GlutaMax^TM^ (Thermo Fisher Scientific, Life Technologies Ltd., Paisley, UK), supplemented with non-essential amino acids (Sigma-Aldrich, Gillingham, UK), 100,000 U/mL penicillin, 100 μg/mL streptomycin and 2% heat-inactivated fetal calf serum (HI-FCS), and 100 μL of each sample dilution was inoculated onto duplicate wells of 80% confluent MDBK cells in 12-well tissue culture plates. Virus was absorbed for 90 min at 37 °C in 5% CO_2_ in air, the inoculum was removed, and the cells were overlaid with medium containing 2% HI-FCS in 1% low gelling temperature agarose (Merck Chemicals Ltd., Gillingham, UK). Following 3 to 4 days of incubation, the agarose was removed, cells were fixed with 80% cold acetone for 15 min and allowed to dry. Cells were blocked with 300 μL 50% Sea Block blocking buffer (Thermo Fisher Scientific, Life Technologies Ltd., Paisley, UK) in PBS for 1 h at room temperature (RT), washed once with PBS containing 0.05% Tween-20 (PBS/Tw) and 300 μL of a mixture of purified monoclonal antibodies (mAb) 19, specific for the RSV F protein [22] and mAb 8, specific for the M2 protein [23], at a final concentration of 1 μg/mL, was added to each well. After incubation for 1 h at RT, plates were washed 3× with PBS/Tw, and bound antibodies to RSV-infected cells were detected by adding 300 µL of 1/2000 HRP-goat anti-mouse immunoglobulin (Ig)G (KPL, Sera Care, Insight Biotechnology Limited, Wembley, UK) to each well. After 1 h at RT, plates were washed 3× with PBS/Tw, and 300 μL of 3-Amino-9-Ethylcarbazole substrate (Merck Chemicals Ltd., Gillingham, UK) was added to each well and incubated for 30 min at RT to allow a red color to develop in BRSV-infected cells. The reaction was stopped by washing in water. Virus titers were expressed as log_10_ ifu (infectious foci units)/mL.

### 2.5. Serology

BRSV-specific serum MDA was analyzed for allocation of calves into treatment groups using a commercial indirect ELISA (SVANOVIR, SVANOVA Boehringer Ingelheim, Uppsala, Sweden), according to the manufacturer’s instructions. BRSV-specific serum and nasal IgA responses were determined using a capture ELISA, as described previously [17]. Levels of BRSV-specific IgG (IgG1 and IgG2) following vaccination were analyzed by ELISAs on 96-well plates coated overnight with an extract of BRSV-infected-Vero cells (BRSV strain 2B, kindly provided by Professor J.A. Melero; Laboratorio de Referencia e Investigación en Virus Respiratorios. Unidad de Biología Viral. Centro Nacional de Microbiología, Instituto de Salud Carlos III) and, in parallel, on plates coated with an extract of non-infected Vero cells (control antigen) in carbonate buffer (pH 9.6). After blocking with 3% bovine seroalbumin (BSA) in PBS, the plates were incubated for 1 h at 37 °C with serum samples diluted 1/200 for detection of IgG1 antibodies or diluted 1/50 for detection of IgG2 antibodies. The bound antibodies were detected using horseradish peroxidase-labelled (HRP) monoclonal anti-bovine IgG1 (EC10, INGENASA) or polyclonal anti-bovine IgG2 (PA1-84661, Thermo Fisher Scientific, Rockford, USA) incubated at 37 °C for 30 min. Washes between consecutive steps were performed with 0.05% Tween 20 in PBS. TMB-MAX (Neogen Corporation, Lexington, KY, USA) and 0.5-M sulfuric acid served as substrate and stopping solution, respectively. The absorbance was measured at 450 nm in a Multiscan Ascent ELISA reader. The results were expressed as SPs calculated as (optical density (OD) serum on BRSV antigen-OD serum on control antigen)/(OD positive control serum on BRSV antigen-OD-positive control serum on control antigen) for IgG1 and IgG2. The cut off was established as a percentage of the SP of positive control serum (20% for IgG1 and 40% for IgG2).

BRSV-neutralizing antibody titers were analyzed by a modification of the method described previously [24]. Briefly, serial dilutions of immune sera were incubated with ~1000 PFU of rBRSV (A51908 strain) expressing GFP [25], in duplicate for 1 h at 37 °C in 5% CO_2_ prior to the addition of 2 × 10^4^ Vero cells/well. Cells were then incubated for 3 days and GFP expression in infected cells was analyzed using a TECAN 200 plate reader at 485 nm and 535 nm absorbance. Neutralizing titers were defined as the reciprocal of the serum dilution giving a 50% reduction in absorbance compared to virus control wells.

BRSV PreF-protein-specific IgG antibody titers were determined by ELISA as described previously [11], using 200 ng of His-tagged PreF protein per well of Pierce^R^ Nickel coated 96-wells plates (Thermo Fisher Scientific, Life Technologies Ltd., Paisley, UK), and 50% Sea Block blocking buffer in PBS. Bound antibodies were detected with HRP-rabbit anti-bovine IgG (Sigma-Aldrich, Gillingham, UK), 1-step^TM^ Ultra TMB ELISA substrate (Thermo Fisher Scientific, Life Technologies Ltd., Paisley, UK), and 2M H_2_SO_4_ as stopping solution. Titers were expressed as log_10_ of the highest dilution of sera giving an OD 1.5× that of the background.

### 2.6. T-cell Responses 

BRSV-specific cell-mediated immune responses were analyzed by IFNγ-ELISpot of γ/δ-depleted PBMCs, stimulated either with heat-inactivated BRSV-infected cell lysate at 10^4.5^ TCID_50_/mL (isolate no. 9402022 Denmark, kindly provided by Pr L E Larsen, University of Copenhagen, Department of Veterinary and Animal Sciences, Frederiksberg C, Denmark) or uninfected cell lysate, as previously described [12].

### 2.7. Histological Analysis of Lung Tissue

Histopathological analyses were performed in a blinded manner on 5-µm sections of tissue samples from the cranial lung lobes, collected at post-mortem examination. The sections were stained with hematoxylin and eosin or Luna’s histochemical stains [26] to demonstrate neutrophils and eosinophils, respectively. The sections were evaluated and morphologically described in a blinded manner by using a Nikon Eclipse Ci-L microscope (Bergman Labora AB, Stockholm, Sweden). The presence and location of neutrophilic infiltrates were graded subjectively by a diagnostic pathologist considering the amount of infiltrating cells and the extent of the section affected. The scores were: – (absent) no neutrophilic infiltrate detected; (+) (insignificant) few scattered solitary neutrophils detected; + (mild) small areas with infiltration of few neutrophils; ++ (moderate) multifocal areas with evident infiltration of neutrophils; +++ (severe) widespread areas with prominent neutrophilic infiltration. The number of eosinophils was evaluated in the sections stained with Luna’s histochemical stain by counting the number of eosinophils in 20 intralesional high power fields (HPF) corresponding to 4.74 mm^2^ (40× objective and 10× ocular with field number 22).

### 2.8. Label-Free Quantitative Mass Spectrometry-Based Proteomics

Proteins in BAL supernatants were analysed by tandem-mass spectrometry as previously described [19,27] with minor modifications. In brief, sample volumes were reduced by using a SpeedVac vacuum concentrator (Thermo Scientific, Bremen, Germany), and samples corresponding to 7 μg of protein were mixed with ammonium bicarbonate, reduced with dithiothreitol (Merck, Stockholm, Sweden), alkylated with iodoacetamide (Merck, Stockholm, Sweden) and in-solution digested by trypsin overnight. The peptides were purified by using Pierce C18 Spin Columns (Thermo Scientific, Bremen, Germany), dried and resolved in 0.1% formic acid. They were thereafter separated by reversed phase liquid chromatography using an EASY-nLC 1000 system, a C18-column and a 90 min linear gradient with acetonitrile. Subsequently, they were nano electrosprayed to a Q Exactive Plus Orbitrap mass spectrometer (Thermo Scientific, Bremen, Germany) and tandem mass spectrometry was performed applying higher-energy collisional dissociation fragmentation on the 10 most intense peaks.

Proteins were identified and quantified using the quantification software MaxQuant 1.5.3.30. [27] and the *Bos taurus* proteome database extracted from UniProt 03/07/2018, with search parameters and criteria for protein identification as detailed previously [19]. A decoy search database including common contaminations and reverse database was used to estimate the identification false discovery rate (FDR). An FDR of 1% was accepted for peptides and protein identification. The results of all fractions were combined to a total label free intensity analysis for each sample. An additional similar search was performed by using a combined fasta file containing sequences for the eosinophil granule major basic protein (*Mus musculus*), eosinophil-derived neurotoxin (*Bos taurus*), eosinophil cationic protein-like isoform X1 (*Bos taurus*), eosinophil peroxidase (*Bos taurus*), eosinophil peroxidase preprotein (*Mus musculus*), and galectin-10 (*Homo sapiens*). Correlations between disease parameters and the relative quantity of detected proteins were performed on proteins detected in at least four animals per group.

### 2.9. Statistical Analysis

Statistical analyses were performed in Minitab version 16, using one-way ANOVA followed by Tukey’s test for normally distributed data, and Kruskal-Wallis test followed by Mann-Whitney test for data which followed a non-normal distribution. The distribution was tested by using Anderson-Darlings test. Data shown are either individual or means with standard deviation (SD). The cut off for statistical significance was set to <0.05. FactoMineR R package [28] was used for principal component analysis (PCA) to get a descriptive analysis of the data and R [29] for the computation of Spearman’s correlations. For statistics of the proteomic data, the LFQ intensities were used as input values.

### 2.10. Ethics Statement 

Experiments were carried out in compliance with French national rules and received the agreement number APAFIS#9903-2017051513402904v6 after approval by the Ethical Committee of Val de Loire, France.

## 3. Results

### 3.1. Virus Replication in Calves Following Intranasal Vaccination with the Live-Attenuated Vaccines

Following i.n. administration, the commercial vaccine (CV) tended to replicate more efficiently than ΔSHrBRSV in the upper respiratory tract of calves, but this difference was not significant on any day post-vaccination (Figure 2). In total, BRSV-RNA was detected by RT-qPCR on three occasions in two/six ΔSHrBRSV-immunized calves, between D6 and D8 pv (mean 97, range 2.5–283 TCID_50_ equivalents), and on eight occasions in 4/6 CV-immunized calves, between D4 and D8 pv (mean 2955, range 2.8 -16408 TCID50 equivalents).

### 3.2. Mild Clinical Reactions Were Observed after Intranasal and Intramuscular Vaccination

Mild adverse clinical signs were observed post-vaccination, within the range of acceptable signs for cattle. Elevated body temperature (defined as >39.5 °C) was recorded on D1 or D2 and lasted for one or two days in four, two, zero, and one calf (calves) vaccinated with ISA61, PreF, ΔSHrBRSV, or the CV, respectively. Body temperature that exceeded 39.9 °C were recorded in two calves injected with ISA61 alone and one in calf immunized with PreF (40–40.1 °C, respectively).

### 3.3. PreF, ΔSHrBRSV and CV Induced Clinical Protection against BRSV Challenge, Three Months after A Single Immunization

Despite the presence of moderate levels of BRSV-specific maternal antibodies at the time of vaccination, PreF, ΔSHrBRSV and the CV induced significant clinical protection against BRSV challenge (Figure 3). PreF-induced protection with the highest significant difference compared to adjuvant alone. The differences between the vaccines were not significant using one-way ANOVA (Figure 3), but PreF induced significantly better clinical protection than the CV on D8 pi by Kruskal-Wallis test adjusted for ties, followed by Mann-Whitney (*p* = 0.037, data not shown).

In line with previous results from this experimental model [15], all controls developed mild to severe clinical signs of respiratory disease. These were first observed D4 to D5 pi, reached a peak at D7 to D8 pi and thereafter declined until termination of the experiment on D13 pi (Figure 3). The disease was characterized by fever, increased respiratory rates, mild-to-severe abdominal and expiratory dyspnea, and enhanced respiratory noise, with or without wheezing. Two controls (calf a and e) showed severe signs including decreased demeanor, fever (≥ 40.7 °C), reduced appetite, cough, nasal discharge, moderate-to-severe abdominal dyspnea, wheezing respiratory noise on auscultation, and increased respiratory rate (≥80 breaths/min). Calf “a” improved spontaneously without medication, whereas calf “e” had persisting clinical signs. On its fourth day of fever (D8 pi), calf “e” was medicated with a nonsteroidal anti-inflammatory drug (meloxicam 0.5 mg/kg subcutaneously, Boehringer Ingelheim Animal Health, Paris, France), following predetermined ethical criteria. The rectal temperature became normal (≤39.5 °C) within 8 h, increased to 39.8 °C on D9 pi and was again normal on D10-D13 pi. Overall, the general condition, respiratory rate and intensity of the respiratory noise of this calf improved, whereas the dyspnea and cough remained until D13 pi. 

In contrast to the controls, all vaccinated animals exhibited clinical protection against BRSV disease. Signs of respiratory disease were absent or were mild and consisted of (one or more of the following): slight nasal discharge, cough, slight increase of respiratory rate, mild abdominal dyspnea and slight wheezing at auscultation. The daily sum of clinical scores differed significantly between controls and calves immunized with any of the three vaccines. The highest significance was detected between PreF-immunized calves and controls (Figure 3). Accordingly, the area under the curve of clinical scores was smallest for calves immunized with PreF (Appendix A).

The peak respiratory rates were significantly lower in both PreF- and ΔSHrBRSV-immunized calves, compared to controls, whereas the peak rectal temperatures were significantly lower in CV-immunized calves compared to controls (0.01 ≤ *p* < 0.05, Table 1).

### 3.4. PreF, ΔSHrBRSV, and CV Induced Protection against Neutrophilic Inflammation Detected in BAL Following BRSV Challenge

Whereas the proportion of granulocytes in BAL was less than 20% in the majority of calves before BRSV challenge (D-1/D-2 pi), there was a large increase in the proportion of granulocytes (up to 60% to 74.5%) in all control calves by D13 pi (Figure 4a). The proportion of granulocytes in control calves at this time was significantly greater than that in calves immunized with any of the vaccines. However, there was a transient increase in the proportion of granulocytes in BAL in CV-immunized calves, on D6/D7 pi, which decreased to pre-challenge levels in all but one calf by D13 pi. In contrast, both PreF- and ΔSHrBRSV-immunized calves had a significantly lower proportion of granulocytes in BAL compared to controls and to CV-immunized calves on D6/D7 pi (Figure 4a).

The data based on cytology were confirmed by tandem-mass spectrometric analyses focused on neutrophil-related proteins in BAL. All BAL samples with a sufficient protein content were analyzed, (*n* = 69). Thirty-seven neutrophil-related proteins were identified and quantified, among a total of 1422 proteins in all samples analyzed. Overall, the proportion of these proteins was highest in the controls on D6/D7 pi, followed by that in the CV-immunized calves on D6/D7 pi and in the controls on D13 pi (Figure 4b). The sum of label-free quantification (LFQ) intensities for the neutrophil-related proteins thus reached the highest values for these calves and occasions, with 1.43 × 10^11^, 1.13 × 10^11^, and 4.66 × 10^10^ intensity units for controls on D6/D7 pi, CV-immunized calves on D6/D7 pi, and controls on D13 pi, respectively. In general, the data therefore indicated that vaccination led to lower means of (LFQ) intensities for the neutrophil-related proteins on D6/7 pi (Figure 4a,b). Compared to the controls, immunized calves had significantly different means of LFQ intensities for several neutrophil-related proteins on D6/D7 pi, and this difference was most pronounced in calves immunized with PreF, followed by ΔSHrBRSV and the CV (Figure 4b). Eosinophil-related proteins were not detected in the searches performed against *Bos taurus* or *Homo sapiens* databases. By using the *Mus musculus* database, the eosinophil cationic protein-like isoform X1 was detected in one PreF-immunized calf, calf l, on D6 pi, and the eosinophil peroxidase preproprotein was identified in one control, calf f, on D13 pi. 

### 3.5. PreF and ΔSHrBRSV Induced Protection against Macroscopic and Microscopic Lung Lesions Following BRSV Challenge

Although the necropsy was performed on D13 pi, which was six days after the peak of the clinical signs (Figure 3), macroscopic lung lesions were observed in all controls and in three CV-vaccinated calves (Figure 5a). Moreover, five/six controls had hemorrhagic and/or enlarged respiratory lymph nodes (calves a, c, d, e, and f), and two/six controls had subpleural emphysema, which agreed with the severity of clinical signs (calf a: left caudal lobe and calf e: left and right cranial and caudal lobes and accessory lobe). In contrast to the CV, both PreF and ΔSHrBRSV induced a significant protection against macroscopic lesions compared to adjuvant alone (Figure 5a).

In agreement with the gross pathology, BAL cytology, and proteomics, histopathological examination revealed an acute to subacute bronchointerstitial pneumonia in controls, with an increased infiltration of neutrophils in the airways, compared to vaccinated calves (Table 2; for examples of the histopathological patterns, see Figure 5b). Furthermore, there were mononuclear inflammatory cells, such as macrophages, lymphocytes and plasma cells, in the interstitial of all calves.

A separate histological analysis was carried out to determine the eosinophilic pulmonary infiltrate. The degree of eosinophil infiltration was measured as the sum of eosinophils in 20 intralesional high power fields (4.74 mm^2^) on D13 pi. Three PreF-immunized calves (i, k, and l) had the highest number of eosinophils in the lung section examined and PreF-immunized calves had significantly higher number of eosinophils than calves immunized with the CV (*p* = 0.029, Table 2).

### 3.6. PreF, ΔSHrBRSV, and the CV Induced Virological Protection in Decreasing Order

In line with earlier experiments performed with this model [15], the kinetics of virus RNA detection spanned from D2 to D13 pi, with the mean peak of viral shedding in the nasal secretions on D6 pi for controls, D5 pi for PreF- and ΔSHrBRSV-immunized calves, and on D4 pi for CV-immunized calves (Figure 6, Appendix A). 

While controls shed large amounts of virus after challenge, PreF-immunized calves were strongly protected against virus replication both in the upper and lower respiratory tract (Figure 6a–c). Viral RNA was detected by RT-qPCR, both in nasal secretions and BAL, in all controls and in all ΔSHrBRSV- and CV-immunized calves, but not in three/six and four/six PreF-immunized calves, in nose and BAL, respectively. Thus, BRSV-RNA was not detected on any occasion in nasal secretions or in BAL of calves i, k, and l (those PreF-immunized calves with the highest eosinophil counts in the lung tissue; Table 2 and Figure 6a–c), nor on any occasion in the BAL of calf g (the PreF-immunized calf with the most neutrophils in lung tissue within its group (Table 2 and Figure 6b,c). 

ΔSHrBRSV and the CV also induced significant virological protection, although not as strong as PreF. The area under the curve of mean nasal virus replication was 153622, 326, 3726, and 31908 TCID_50_ equivalent units for calves injected with ISA61 or vaccinated with PreF, ΔSHrBRSV, or CV, respectively (Appendix A). The level of viral RNA was significantly lower in PreF- compared to ΔSHrBRSV-immunized calves on D4 and D5 pi (Figure 6a). 

In agreement with the results obtained in the upper respiratory tract, all vaccines induced a significant reduction in the level of BRSV-RNA in BAL on D6/D7 pi and D13 pi compared to controls (Figure 6b). Moreover, the quantity of BRSV-RNA in BAL was significantly lower in PreF- compared to that in CV-immunized calves on D6/D7 pi (Figure 6b). The data based on viral RNA in the lower respiratory tract were confirmed by virus isolation from BAL. All vaccinated calves were significantly virologically protected in the lower respiratory tract on D6/D7 pi, compared to controls (Figure 6c). Whereas BRSV was isolated from BAL of five/five controls (calves b-f), it was only isolated from 1/18 of the vaccinated calves (a calf vaccinated with the CV: calf t, Figure 6c). This particular calf had the highest level of BRSV-RNA in BAL compared to the other CV-immunized calves (Figure 6b).

The higher level of viral replication in controls was confirmed by tandem mass spectrometry, since BRSV proteins were detected in BAL only from controls and only on D6/D7 pi (calf a (n.a.); calf b (BRSV P protein); calves c; d; e; and f (BRSV M-, BRSV M21-, BRSV N-, and BRSV P protein), data not shown).

### 3.7. In Contrast to ΔSHrBRSV and the CV, PreF Induced BRSV-Specific Humoral Responses Pre-Challenge in Most Animals and Primed for Neutralising Antibodies Post-Challenge

Calves were allocated to the different treatment groups based on breed, age and BRSV-specific MDA titers at two to three weeks of age and there was therefore no significant difference in BRSV-specific MDA in serum between calves in the different groups, on the day of vaccination (the mean titers were 129, 150, 108, and 125 for controls and calves immunized with PreF, ΔSHrBRSV, and the CV, respectively, Figure 7a). By D56 pv, only PreF-vaccinated calves had developed BRSV-specific serum IgG1 and IgG2 antibodies (Figure 7b,c), and by D84 pv, these calves had significantly higher titers of BRSV-specific serum IgG than calves immunized with ΔSHrBRSV or the CV (Figure 7a). Moreover, the BRSV-specific IgG and IgG2 responses were stronger in PreF-vaccinated calves than in either ΔSHrBRSV-vaccinated or CV-vaccinated calves post-challenge (Figure 7a,c).

The BRSV-neutralizing antibody titers did not differ significantly between calves on the day of vaccination. Despite the presence of MDA, PreF induced BRSV-neutralizing antibodies in four/six calves by D56 pv (Figure 8). The BRSV-neutralizing antibody titers were significantly higher in PreF-vaccinated calves than in ΔSHrBRSV- or CV-vaccinated calves on D56 pv and only PreF primed for a significantly higher BRSV-neutralizing antibody response than control calves, on D105 pv, 13 days after BRSV challenge (Figure 8).

On D84 pv, BRSV PreF-specific antibody titers greater than 3.5 log_10_ were detected in four/six calves immunized with PreF and in one/six calves immunized with ΔSHrBRSV (Figure 9). PreF induced the strongest BRSV-PreF-specific serum IgG responses following challenge (D13 pi). 

### 3.8. None of the Vaccines Primed for IgA or Circulating BRSV-Specific IFNγ Expressing T Cells

Following BRSV challenge, the PreF-immunized calves developed significantly lower levels of BRSV-specific IgA in serum compared to controls or ΔSHrBRSV-immunized calves (Figure 10a). In fact, none of the vaccines appeared to prime for an anamnestic nasal or systemic IgA response at D13 post-challenge, or induce BRSV-specific serum IgA with long duration following vaccination (Figure 10a,b). Circulating BRSV-specific and IFNγ expressing memory T cells were detected by ELISpot in two CV-immunized calves D84 pv and in several calves in all groups D13 pi but without any significant differences between calves immunized with different vaccines or between immunized calves and controls (Figure 10c).

### 3.9. Immunological Markers before and after Challenge Correlated with Protection against Challenge

With the exception of one ΔSHrBRSV-immunized calf (calf m), the pattern of the different immune and disease parameters that were analyzed were relatively homogenous within each group of calves (Figure 11a). When comparing the confidence ellipses, the three groups of vaccinated calves were distinct from the group of control calves, and the group of PreF vaccinated calves was distinct from the two other vaccinated groups (Figure 11a). The confidence ellipses from the two live-attenuated i.n. vaccines overlapped in the dimensions one versus two of the identified PCA factors, supporting more closeness between the live intranasal vaccines than with the subunit PreF-vaccine administrated by the parenteral route. When analyzing data from all calves simultaneously, a trend for an opposite direction (and thus probable negative correlation) was identified between the humoral response detected after challenge (_post, Figure 11b) except for IgA in serum, and the parameters of infection and clinical disease (hereafter called disease parameters). All immune parameters obtained before challenge (_pre, Figure 11b) were in the opposite direction of the disease parameter PMN_BAL_D6D7 that measured the inflammatory cellular recruitment in BAL at the peak of disease (Figure 11b).

Correlation analyses were performed separately for each group of six vaccinated calves, between immune parameters obtained pre-challenge and disease parameters obtained post-challenge. Thus, this analysis targeted parameters that contributed to the prediction of protection against challenge at the individual level within a group of vaccinated animals. In PreF-vaccinated calves, BRSV-specific serum IgG pre-challenge correlated negatively with the area under the curve of BRSV RNA in nasal secretions (Spearman rho −0.943, *p* = 0.01) and PMN in BAL post-challenge (Spearman rho −0.83, *p* = 0.04). In ΔSH-vaccinated calves, BRSV-specific serum IgG2 pre-challenge correlated negatively with eosinophils in lung tissue post-challenge (Spearman rho −0.83, *p* = 0.04). In CV-vaccinated calves, no significant correlations were identified between pre-challenge immune parameters and post-challenge disease parameters.

### 3.10. Several Proteins in BAL Obtained before Challenge Correlated with Protection against Challenge

The entire BAL proteomes obtained by tandem mass spectrometry three months after immunization and before challenge (D91/D92 pv; D-2/D-1 pi) were analyzed for correlations with disease parameters within each group of vaccinated calves and controls, and proteins with the most negative correlations with disease were identified. These proteins differed in calves immunized with different vaccines and even in calves immunized with the intranasal vaccines (Table 3). 

In PreF-vaccinated calves, an immunoglobulin-like protein (A5D7Q2) and Whey-Acidic Protein (WAP) four-disulfide core domain protein 2 (WFDC2) correlated negatively with five and four disease parameters, respectively (Table 3). In ΔSHrBRSV-immunized calves, alkaline phosphatase (ALPL) and glucose-6-phosphopatase isomerase (GPI) correlated negatively with five and three disease parameters, respectively (Table 3). In CV-immunized calves, secretoglobin family 1A member 1 (SCGB1A1) was the only protein that correlated negatively with more than one disease parameter (Table 3).

Overall, the identified proteins were associated with immunoglobulins (immunoglobulin like (A5D7Q2), Ig V-set (G5E5H2), immunoglobulin light chain lambda (IGL), polymeric immunoglobulin receptor (PIGR)); metal-binding or pathogen-killing (aldehyde oxidase 3L1 (AOX2), lysozyme C (LYS)); monocytes, neutrophils, or acute phase responses ((S100A8, S100A12, secretoglobin (SCGB2A2 or SCGB1A1), alpha-1-acid glycoprotein (ORM1), peptidoglycan recognition protein (PGLYRP1)); reduction of inflammation (alkaline phosphatase (ALPL)); epithelial-to-mesenchymal transformation; cell migration or lymphocyte homing (WAP four-disulfide core domain 2 (WFDC2), CD44) and metabolism (Glucose-6-phosphopatase isomerase (GPI), Rho GDP-dissociation inhibitor 2 (ARHGD), L-lactate dehydrogenase B chain (LDHB), and 6-phosphogluconolactonase (PGLS).

## 4. Discussion

While safety and efficacy are the primary general goals in HRSV- and BRSV-vaccine development, the duration of protection induced by a single vaccination in the presence of virus-specific MDA would be very advantageous. Indeed, in both humans and cattle, this would make vaccination more robust and would enable vaccine usage at a large scale in different societal, economical and geographical contexts. Based on others and our previous results [11,12,13,14,30], bovine PreF and ΔSHrBRSV both appear promising candidates, either as new vaccines against BRSV in cattle, or as proof of concept for vaccines against HRSV in babies. In the present study, a BRSV calf model was used to compare the efficacy and the safety of a single administration of each of these vaccines and a commercial BRSV vaccine, three months post-vaccination. The results demonstrated that PreF in adjuvant, administrated once by the i.m. route to calves with BRSV-specific MDA, induced strong clinical and virological protection, which lasted at least 92 days. These data extend previous results obtained with PreF administered to seronegative or seropositive calves, with a shorter interval between vaccination and challenge, [11,12] and compares this subunit parenteral vaccine with two live-attenuated intranasal vaccines.

Significant clinical protection was induced by all three vaccines compared to adjuvant alone. No noticeable clinical signs were observed in calves immunized with PreF and, although not significantly different, less clinical signs were observed in calves immunized with ΔSHrBRSV than the CV. The clinical observations were in agreement with the degree of inflammation based on cytology and the proteomic data, as well as with the pathological observations. On D6/D7 pi, at the acute phase of the disease, both PreF- and ΔSHrBRSV-immunized calves had lower proportions of granulocytes and neutrophil-related proteins in BAL than controls and calves immunized with the CV, probably as a result of a better virological protection. Although the necropsies were performed on D13 pi, nearly a week after the acute clinical signs, macroscopic lesions were exclusively recorded in the lungs of controls (in six/six calves) and in calves immunized with the CV (in three/six calves). The lesions observed in the CV-vaccinated calves were not as severe, nor as acute as in the controls, since microscopically, only the controls had an extensive infiltration of neutrophils in these lesions, consistent with broncho-interstitial pneumonia. The macroscopic lesions in the calves immunized with the commercial vaccine probably derived from a resolved inflammation following BRSV challenge, since the three CV-immunized calves with lesions (calves t, v, and x) had most BRSV RNA in nasal secretions and BAL within their group.

Overall, the virological data were in agreement with the pathology, the clinical observations and the pulmonary inflammatory results. Compared to controls, the longitudinal detection of BRSV-RNA in nasal secretions was 470, 40, and 5 times lower in calves immunized with PreF, ΔSHrBRSV, and the CV, respectively. Live BRSV was isolated from the BAL of all controls from which BAL was obtained on D6/7 pi (5/6), from only one out of six CV-vaccinated calves (calf t), but not from any of the PreF- or ΔSHrBRSV-immunized calves. Whereas BRSV-RNA was detected in both the upper and lower respiratory tract of all controls and all calves immunized with ΔSHrBRSV or the CV, only very low amounts were detected in the upper and lower respiratory tract of three and two PreF-immunized calves, respectively. Some of the variation that was observed among calves immunized with the same vaccine might be related to the level of BRSV-neutralizing MDA at immunization. Nevertheless, the virological data confirmed that PreF induced the strongest virological protection.

Following immunization, no severe adverse clinical signs were observed. Mild hyperthermia was detected for one or two days, mainly following i.m. administration of adjuvant alone or PreF, but also in one calf immunized i.n. with the CV. These observations were in line with previous vaccine evaluations in cattle, in which water-in-oil adjuvants were used [12,14]. Vaccine virus RNA was detected in nasal secretions of some calves after immunization with either ΔSHrBRSV or the CV, but in higher levels and in a higher proportion of animals in CV- than in ΔSHrBRSV-immunized calves. These results are in agreement with previous reports [13,31], and indicate that ΔSHrBRSV is at least as well attenuated as the CV and support our previous observations that ΔSHrBRSV probably does not spread from vaccinated animals [14]. Despite the lower level of replication, ΔSHrBRSV tended to induce a stronger virological and clinical protection against challenge than the CV.

To enable detection of any immunopathological events in vaccinated calves after infection with virulent BRSV, the calves were monitored for 13 days after challenge. According to clinical observations, gross pathology, the general histopathological pattern and proteomic data on BAL, no adverse immunopathological reactions occurred. However, three PreF-immunized calves with the strongest virological protection (calf i, k, and l), had more eosinophils in lung tissue on D13 pi than the rest of the calves. Although no signs of enhanced disease or other inflammatory markers were observed in these calves, this finding is in line with results on a single PreF-immunized calf in a previous study [12] and require attention. The presence of eosinophils has been a focus of RSV vaccine-induced immunopathology [32,33]. However, in contrast to inactivated vaccines that induced eosinophil responses and were associated with enhanced disease [32,34,35], PreF induced strong BRSV neutralizing responses in serum. Furthermore, since suboptimal doses of HRSV pre- and post-F primed for enhanced immunopathology in mice and cotton rats [36,37], immunization with 10 times less PreF was tested in a separate calf trial and did not induce any exaggerated inflammation following BRSV challenge (Taylor et al. personal communication). Taken together, our data indicate that the PreF vaccine is sufficiently safe to be used in calves with maternal antibodies and supports the suggestions of others [38], who claim that lung eosinophils do not contribute to RSV vaccine-enhanced disease. Nevertheless, the triggering factors for the recruitment of eosinophils and the consequences of a possible activation of these cells should be further investigated.

A long-lasting protection induced by active immunization is desired for BRSV vaccine development. Although passive immunity protects against severe clinical signs of disease [39,40,41], the heterogeneity of colostrum antibody uptake and the decay of MDA over time make a strategy that only relies on colostrum suboptimal to control BRSV. Since calves may be infected from a young age [33,42], another strategy has been active immunization as soon as possible after birth. However, BRSV-specific MDA may then recognize the vaccine antigen and impair immune responses to vaccination. Maternally derived antibodies can be present during the first seven months of the life of the calf [33,39,43]. Several approaches to overcome the negative effect of MDA on vaccination have been presented [44] and some have been applied in cattle in the past, such as those based on repeated parenteral injection [45], use of a powerful adjuvant [12,14,17,46], intranasal immunization to stimulate mucosal immunity [47] and heterologous boosting to prime different arms of the adaptive immune response (Taylor et al. personal observations). Vaccination by the parenteral route usually necessitates two doses with three to four weeks interval to induce clinical and partial virological protection three to four weeks later. To obtain a faster and more effective protection, live vaccines have been administrated by the intranasal route or adjuvanted and administrated by the parenteral route to induce a mucosal and systemic protective immunity [6,14,40,46,47,48]. However, the duration of protection still appears limited to three months [8,46] (JFV personal observations) and a further vaccine boost seems to be necessary for some vaccines after two–four months, to prolong the clinical and the viral protection [8,33,46]. 

In the present study, only PreF induced a significant increase in BRSV-specific serum IgG in calves with MDA in response to vaccination. In agreement with previous results [12], IgA did not seem to contribute to the protection induced by PreF, and PreF did not appear to induce long-term, detectable circulating memory, IFNγ-producing T cell responses. Therefore, the protection induced by PreF appears to be linked to serum BRSV-specific IgG and neutralizing antibodies. The intranasally administrated vaccines, ΔSHrBRSV and the CV, did not induce significant humoral responses until after BRSV challenge, when vaccine-induced anamnestic IgG responses occurred. Nevertheless, BRSV-neutralizing serum antibodies were detected in serum of one ΔSHrBRSV-immunized calf (calf m), as previously observed after ΔSHrBRSV-immunization of calves without, but not with BRSV-specific MDA [13,14]. Indeed, this calf had the lowest level of MDA at immunization within this group (data not shown). The above data have consequences for our understanding of the protective immune response induced by BRSV vaccines several months after initial vaccination. Whereas anamnestic mucosal IgA was considered an important correlate of vaccine-induced protection two to five weeks after immunization of calves with BRSV-specific MDA [14,17], the present data indicate this immune effector contributes less to protection over time. Three months after vaccination, the rapid development of PreF specific, neutralizing serum antibody responses, following BRSV challenge, appeared important, even in the protection induced by ΔSHrBRSV and the CV.

The ability of PreF to induce neutralizing antibodies in calves with MDA may be due to: (i) the effect of the adjuvant, as previously observed [14,17,49]; (ii) differences in the antibody repertoire between the mother, which had been previously vaccinated by the parenteral route with a live BRSV vaccine, and calves, as hypothesized previously [12,44] and described in humans [50]; (iii) the BRSV MDA were mainly directed against antigenic sites on post-F and very little against those present only on PreF (Sites 0 and V) or (iv) the small size of the antigen, which prevented inhibition of B-cell activation [44]. However, neutralizing antibodies induced by BRSV infection are mainly directed against pre-F rather than post-F [30]. Therefore, BRSV-specific MDA most probably contain at least some antibodies against antigenic sites 0 and V. In conclusion, the long-lasting efficacy of the PreF vaccine is probably based on the combination of an efficient adjuvant, an antigen that can induce a protective effect and differences in the antibody repertoires between calves and adults.

The identification of poorly protected animals or humans prior to natural or experimental virus challenge is needed to tailor effective vaccination programs. Overall, it appears that at the level of a group of calves, it is possible to predict the protection induced by PreF by using seroneutralizing tests or ELISAs to measure the level of antibodies specific against PreF, three months after immunization. This is strengthened by the fact that among the six PreF-vaccinated calves, there was a significant negative correlation between BRSV-specific serum IgG pre challenge and nasal BRSV as well as PMN cells in BAL post-challenge. This correlation is likely only applicable to PreF-immunized calves, since the total BRSV-specific serum IgG of these calves targeted PreF and, thus, probably mainly consisted of functional antibodies, including neutralizing antibodies detectable in at least four out of six calves at D84 pv.

Similarly, among PreF-vaccinated calves, the relative quantity of immunoglobulin-like protein A5D7Q2 in BAL pre challenge correlated negatively with five different disease parameters post-challenge. The UniProt ID A5D7Q2 stands for an uncharacterized protein that in theory could represent BRSV-PreF-specific antibodies. However, A5D7Q2 has in previous publications been associated with the TAP binding protein, which is involved in antigen processing and assembly of MHC class I molecules [51,52,53]. Pre-challenge levels of the whey acidic protein (WAP) four-disulfide core domain protein 2 (WFDC2) additionally correlated negatively with multiple disease parameters after challenge in PreF-immunized calves and controls. This is a proteinase inhibitor that is overexpressed in neutrophil-based inflammatory disorders, such as cystic fibrosis. It is believed that WFDC2 protects against proteolytic enzymes released by inflammatory cells [54], and/or proteolytic microbial virulence factors [55]. Although it is unlikely that the PreF-antigen directly induced WFDC2 expression, it is possible that PreF-immunization skewed the calf’s innate responses towards protection against infection and exaggerated inflammation. Anti-inflammatory functions have also been associated with the tissue non-specific alkaline phosphatase ALPL [56], which correlated with protection in ΔSHrBRSV-immunized calves. Neutrophils produce ALPL as an autoregulatory mechanism in response to inflammatory stimuli, to dampen their own production of proinflammatory cytokines such as IL-6 [57]. This protein is used to predict asthma exacerbations in humans and is increased in sputum of humans with neutrophilic asthma [58]. Live-attenuated BRSV vaccines were previously shown to induce BRSV-specific Th17 responses, which might lead to IL8 production and more rapid neutrophil responses upon BRSV infection [59]. One could speculate that both specific and non-specific Th17 responses were induced by ΔSHrBRSV, which skewed the innate respiratory immunity to neutrophilic responses and consequently an active neutrophil autoregulation. The strong neutrophilic response detected D6/7 pi in calves that had been vaccinated with CV may be due to lack of protection against virus replication and/or reduced neutrophil autoregulation.

In summary, we identified markers that enabled prediction of protection three months after a single immunization of calves in the presence of BRSV-specific MDA At the group level in PreF-immunized calves, serum immunoglobulins such as BRSV neutralizing antibodies, BRSV- and BRSV-PreF-specific IgG enabled prediction of long-term protection. Among individual PreF-immunized calves, higher levels of the uncharacterized immunoglobulin-like protein A5D7Q2 and WFDC2 correlated with protection. Although calves immunized with ΔSHrBRSV or the CV did not develop a BRSV-specific antibody response following vaccination, ΔSHrBRSV-immunized calves that expressed higher levels of ALPL in BAL before challenge appeared to be better protected against BRSV. Overall, both ΔSHrBRSV- and CV-immunized animals were well primed and developed significant BRSV- and BRSV-PreF-specific serum IgG after challenge. To enable prediction of protective antibody responses, in vitro stimulation of memory B cells might be required.

## 5. Conclusions

Taken altogether and to the authors’ knowledge, this study is the first to demonstrate that a one-shot subunit vaccine administrated by the intramuscular route to young calves having specific maternal antibodies can induce clinical, pathological, and virological protection against BRSV for at least three months. This long duration of virological protection is a major breakthrough that will make the implementation of vaccination easier and make this type of vaccine attractive to use as a control tool to stop the circulation of BRSV. Furthermore, if administered before grouping of animals, the duration of protection covers the most critical period e.g., in feedlot production. Although to a slightly lesser extent than PreF, a single intranasal vaccination with ΔSHrBRSV also induced significant protection against BRSV, which was greater than that induced by the CV for protection against pulmonary inflammation. Therefore, PreF and ΔSHrBRSV appear to be competitive alternatives to commercial vaccines already present on the market.

## Figures and Tables

**Figure 1 vaccines-09-00236-f001:**
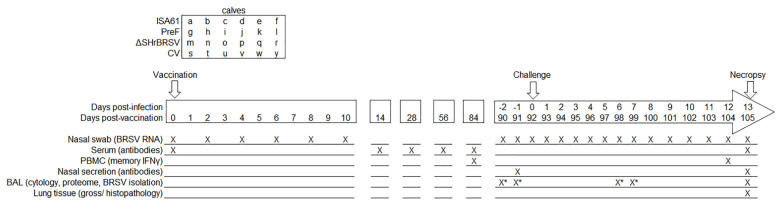
Experiment timeline. Calves were vaccinated on day (D)0 post-vaccination (pv), with either (i) Pre-fusion F adjuvanted by ISA 61 VG (PreF) intramuscularly (i.m.), (ii) ΔSHrBRSV (ΔSHrBRSV) intranasally (i.n.), or (iii) a commercial vaccine (CV) i.n., or were injected i.m. with PBS in ISA 61 VG alone (ISA61) (Vaccination, arrow). Three months later, on D92 pv and D0 post-infection (pi), all calves were challenged with virulent BRSV (Challenge, arrow). The calves were examined daily, and clinical signs of disease were scored from D90 pv/ D-2 pi until D13 pi, when necropsies were performed, and lung tissue was collected for histopathological analyses. Throughout the experiment, samples were collected as indicated in the figure, to analyze antibodies in serum and nasal secretions, BRSV-specific interferon (IFN)γ responses in peripheral blood mononuclear cells (PBMC), live BRSV and BRSV-RNA in nasal secretions and bronchoalveolar lavage (BAL), as well as the proportion of granulocytes and neutrophil-related proteins in BAL at D-2/D-1 and D6/D7 pi.

**Figure 2 vaccines-09-00236-f002:**
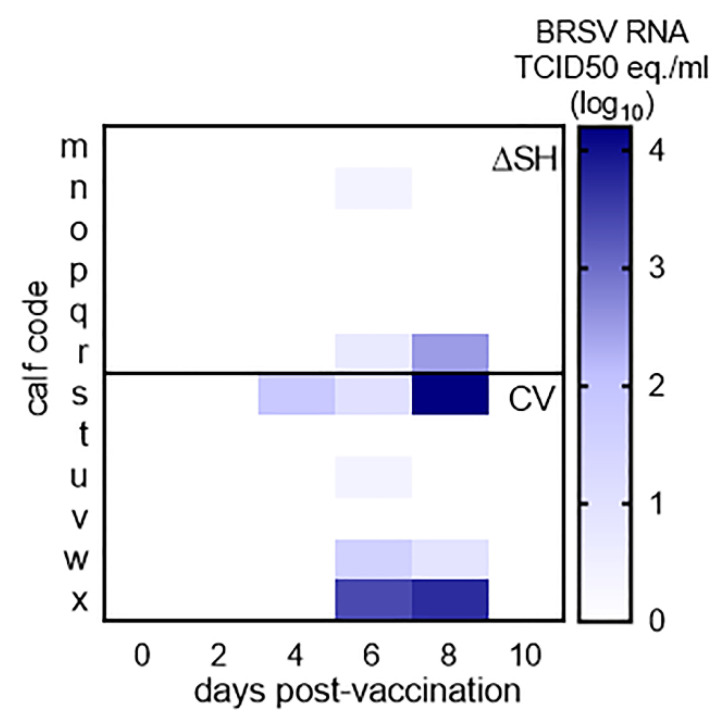
Nasal replication of BRSV vaccine viruses. Calves were vaccinated intranasally on day (D)0 post-vaccination, with either 5 × 10^6^ plaque forming units of ΔSHrBRSV (ΔSH) or one dose of a commercial live-attenuated vaccine (CV). The vaccine-virus shedding was monitored using reverse transcriptase quantitative PCR on the BRSV F-gene after total RNA extraction from nasal swabs. Values are expressed as 50% tissue culture infective dose (TCID_50_) equivalents, calculated from a standard dilution series of virus with a known TCID_50_. The difference was not significantly different.

**Figure 3 vaccines-09-00236-f003:**
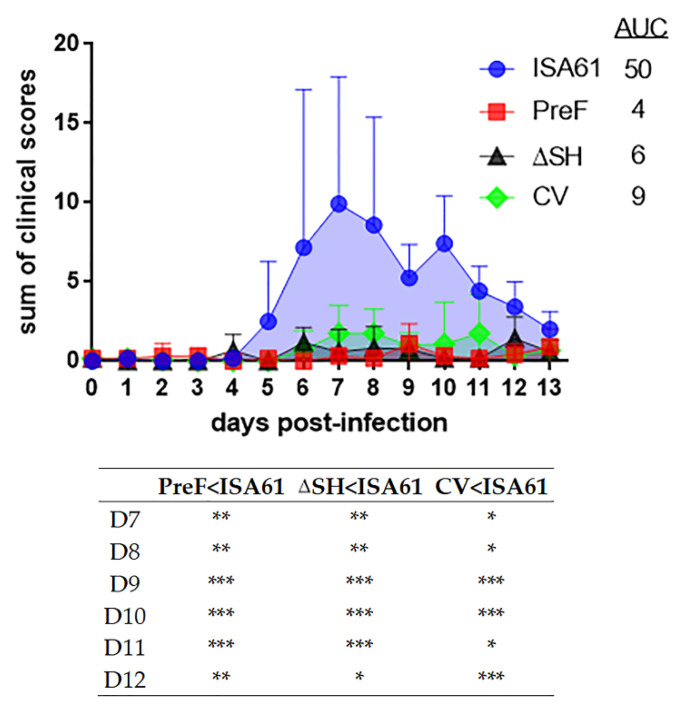
Vaccination protects against clinical signs of bovine respiratory syncytial virus (BRSV). Calves were vaccinated with either (i) Pre-fusion F (PreF) adjuvanted by ISA 61 VG intramuscularly (i.m.), (ii) ΔSHrBRSV (ΔSH) intranasally (i.n.), or (iii) a commercial vaccine (CV) i.n., or were injected with ISA 61 VG alone (ISA61) i.m. and challenged with virulent BRSV, 92 days (D) later on post-infection (p.i.) D0 (D0 pi). The calves were examined daily, and the severity of clinical signs of disease was scored until euthanization on D13 pi. The means of the daily sum of clinical scores are expressed, and the upward deflection lines illustrates the standard deviation. The area under curve (AUC) represents the accumulated means of clinical scores and significant differences between treatment groups are indicated by asterisks (*p* < 0.05 (*), *p* < 0.01 (**), and *p* < 0.001 (***). One-way analysis of variance (two-sided) and Tukey’s pairwise comparisons).

**Figure 4 vaccines-09-00236-f004:**
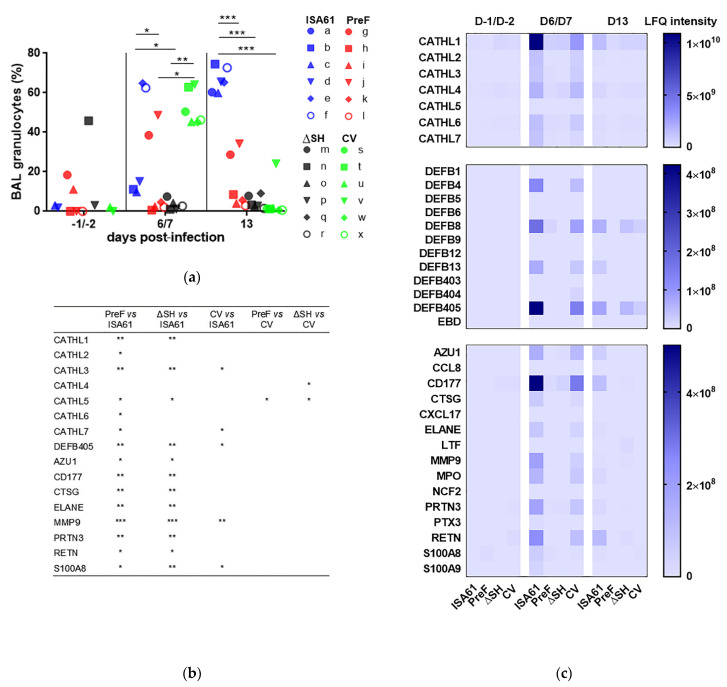
Vaccination reduces polymorphonuclear granulocytes and neutrophil-related proteins in bronchoalveolar lavage (BAL) of calves following bovine respiratory syncytial virus (BRSV) challenge. Calves were vaccinated with either (i) Pre-fusion F (PreF) adjuvanted by ISA 61 VG intramuscularly (i.m.), (ii) ΔSHrBRSV (ΔSH) intranasally (i.n.), or (iii) a commercial vaccine (CV) i.n., or were injected with ISA 61 VG i.m. and were challenged with virulent BRSV, 3 months later. Bronchoalveolar lavages were obtained once before challenge (day (D)-1 or D-2 post-infection, pi) and twice after challenge D6 or D7 and D13 pi. (**a**) Proportion of granulocytes in BAL estimated by cytology. (**b**) Significant differences in the relative quantity of neutrophil-related proteins in BAL D6 or D7 (determined by label-free quantitative mass spectrometry-based proteomics expressed as intensity units), indicated by asterisks (*p* < 0.05 (*), *p* < 0.01 (**), and *p* < 0.001 (***). One-way analysis of variance (two-sided) and Tukey’s pairwise comparisons. (**c**) Illustration of the relative quantity of neutrophil-related proteins described in (**b**), expressed as label-free quantitation (LFQ) intensity.

**Figure 5 vaccines-09-00236-f005:**
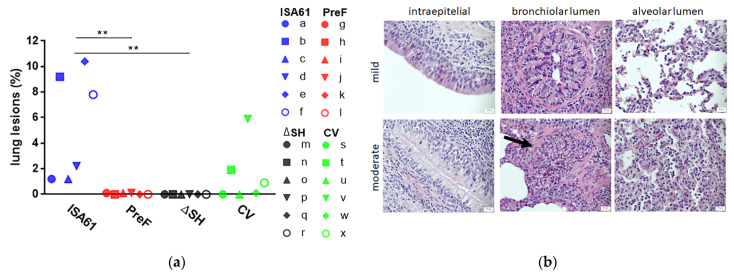
Vaccination reduces the extent of lung lesions following bovine respiratory syncytial virus (BRSV) challenge. Calves were vaccinated with either (i) Pre-fusion F (PreF) adjuvanted by ISA 61 VG intramuscularly (i.m.), (ii) ΔSHrBRSV (ΔSH) intranasally (i.n.), or (iii) a commercial vaccine (CV) i.n., or were injected with ISA 61 VG i.m. and were challenged with virulent BRSV, 3 months later. Thirteen days post-infection, all calves were euthanized, and the lungs were examined for macroscopic lung lesions by observation and palpation. Formalin-fixed tissue samples from the cranial lobe in the right lung were analyzed for identification of histopathological changes, such as granulocytic infiltration. (**a**) The proportion of lung surface with macroscopic lesions, calculated using Image J (free software, version 1.52a) and lung chart drawings. Significant differences between treatment groups are indicated by asterisks, *p* < 0.01 (**). One-way analysis of variance (two-sided) and Tukey’s pairwise comparisons. (**b**) Examples of sections with different neutrophil grading, ×400 magnification. First row: mild lesions, +, from left to right calf number c, f, and d. Bottom row: moderate, ++, from left to right calf number g, d, and b. Arrow indicates a bronchiole.

**Figure 6 vaccines-09-00236-f006:**
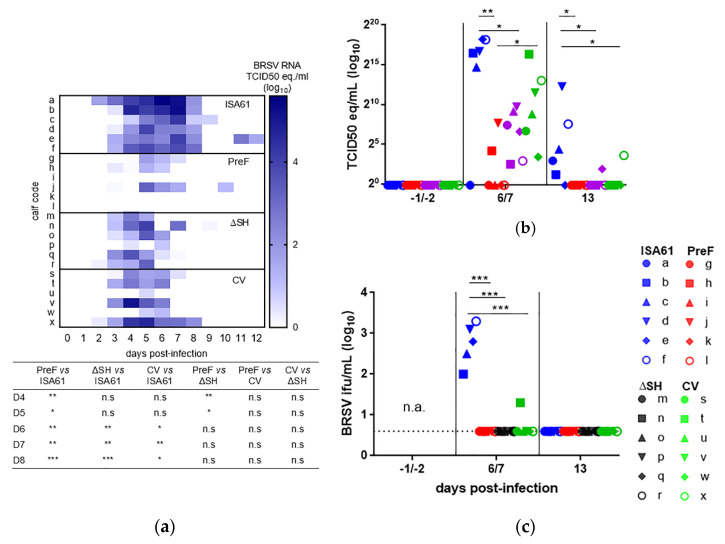
Vaccination reduces virus load in the upper and lower respiratory tract of calves following bovine respiratory syncytial virus (BRSV) challenge. Calves were vaccinated with either (i) Pre-fusion F (PreF) adjuvanted by ISA 61 VG intramuscularly (i.m.), (ii) ΔSHrBRSV (ΔSH) intranasally (i.n.), or (iii) a commercial vaccine (CV) i.n., or were injected with ISA 61 VG i.m. and were challenged with virulent BRSV, 3 months later. Virus replication was monitored using real-time reverse transcriptase quantitative PCR (RT-qPCR) on the BRSV F gene after total RNA extraction. BRSV-RNA is expressed as 50% tissue culture infective dose (TCID_50_) equivalent, calculated from a standard dilution series of virus with a known TCID_50_. (**a**) Virus replication detected in nasal secretions by RT-qPCR. (**b**) BRSV-RNA detected in bronchoalveolar lavage (BAL) by RT-qPCR (**c**) BRSV detection by virus isolation from BAL expressed as log10 infectious foci units (ifu)/mL (dotted line = limit of detection). Significant differences between treatment groups are indicated by asterisks (*p* < 0.05 (*), *p* < 0.01 (**), and *p* < 0.001 (***), Kruskal-Wallis test followed by Mann-Whitney test and one-way analysis of variance (two-sided), followed by Tukey’s pairwise comparisons).

**Figure 7 vaccines-09-00236-f007:**
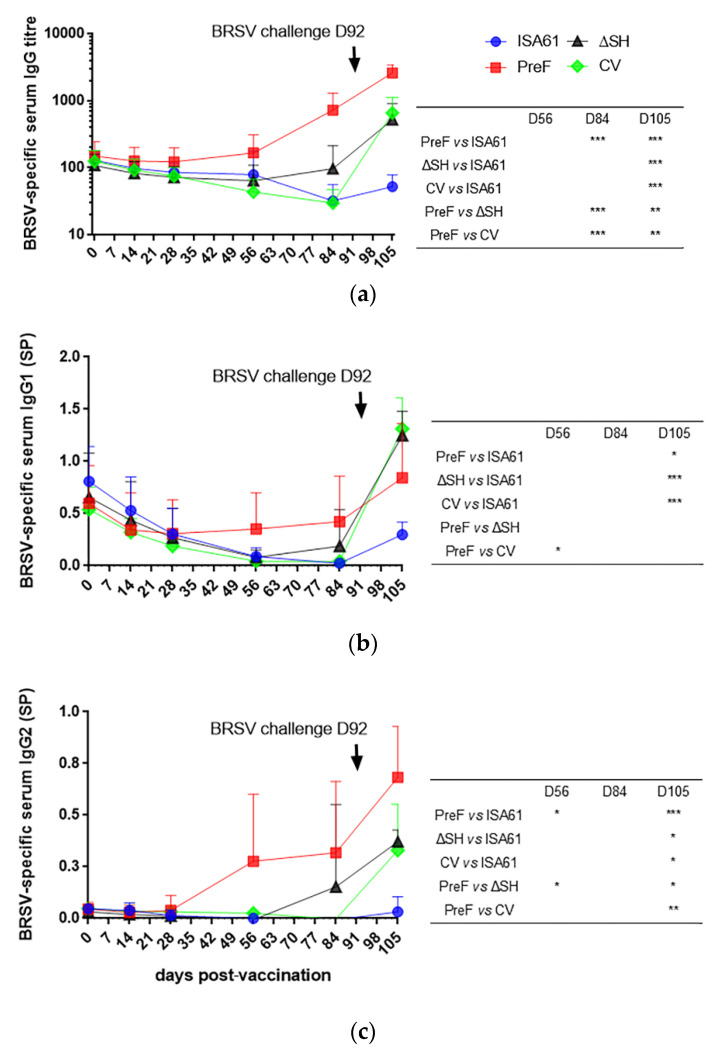
Bovine respiratory syncytial virus (BRSV)-specific serum immunoglobulin (Ig)G antibodies, as determined by enzyme-linked immunosorbent assay (ELISA), in calves before and after immunization on day (D)0 and subsequent BRSV challenge with virulent BRSV on D92 post-vaccination. Calves were vaccinated with either (i) Pre-fusion F (PreF) adjuvanted by ISA 61 VG intramuscularly (i.m.), (ii) ΔSHrBRSV (ΔSH) intranasally (i.n.), or (iii) a commercial vaccine (CV) i.n., or were injected with ISA 61 VG i.m. (**a**) BRSV-specific serum IgG titers, (**b**) BRSV-specific serum IgG1, and (**c**) BRSV-specific serum IgG2, expressed as Sample-to-Positive (SP) and obtained as described in Section 2.5. The upward deflection lines illustrate the standard deviation, and the significant differences between treatment groups are indicated by asterisks (*p* < 0.05 (*), *p* < 0.01 (**), and *p* < 0.001 (***). One-way analysis of variance (two-sided) and Tukey’s pairwise comparisons.

**Figure 8 vaccines-09-00236-f008:**
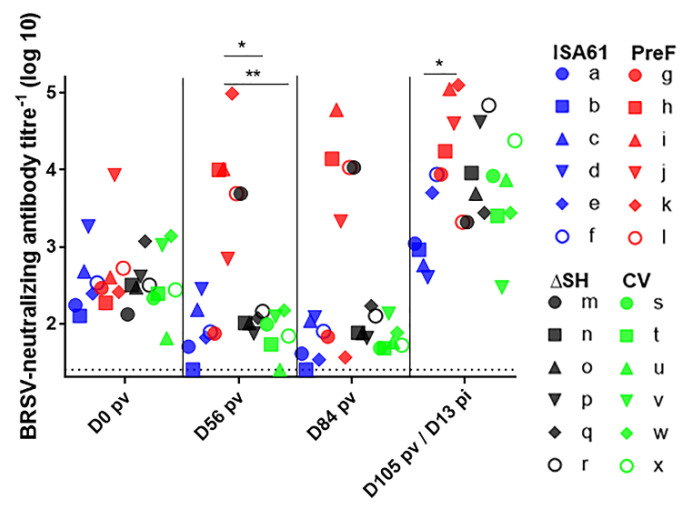
Vaccination with PreF induced bovine respiratory syncytial virus (BRSV)-neutralizing antibodies in calves with maternally derived antibodies. Calves were vaccinated with either (i) Pre-fusion F (PreF) adjuvanted by ISA 61 VG intramuscularly (i.m.), (ii) ΔSHrBRSV (ΔSH) intranasally (i.n.), or (iii) a commercial vaccine (CV) i.n., or were injected with ISA 61 VG i.m. and were challenged with virulent BRSV, 3 months later. The BRSV-neutralizing antibody titers were determined in serum collected three times before infection (day (D)0, D56 and D84 post-vaccination (pv) and D105 pv/D13 post-infection (D13 pi) (dotted line = limit of detection). Significant differences between treatment groups are indicated by asterisks (*p* < 0.05 (*), *p* < 0.01 (**), and *p* < 0.001 (***). One-way analysis of variance (two-sided) and Tukey’s pairwise comparisons.

**Figure 9 vaccines-09-00236-f009:**
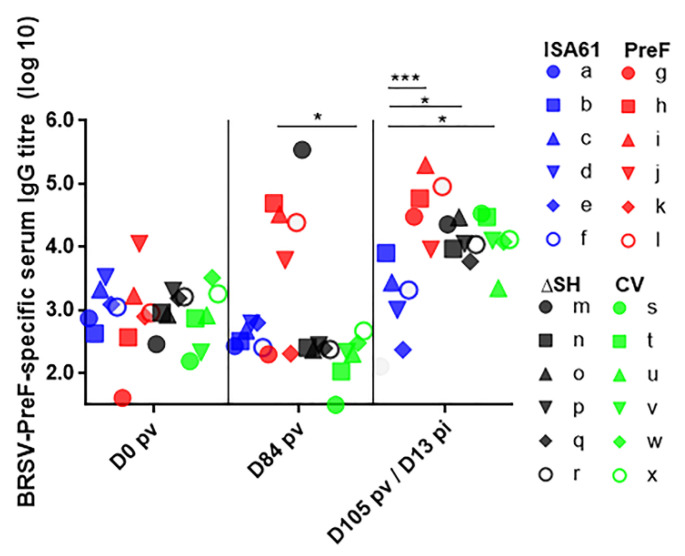
Vaccination with PreF induced bovine respiratory syncytial virus (BRSV)-pre-F-specific immunoglobulin (Ig)G serum antibodies. Calves were vaccinated with either (i) Pre-fusion F (PreF) adjuvanted by ISA 61 VG intramuscularly (i.m.), (ii) ΔSHrBRSV (ΔSH) intranasally (i.n.), or (iii) a commercial vaccine (CV) i.n., or were injected with ISA 61 VG i.m. and were challenged with virulent BRSV, 3 months later. BRSV-PreF-specific serum IgG titers were determined by enzyme-linked immunosorbent assay (ELISA) in serum collected twice before challenge (day (D)0 and D84 post-vaccination, pv) and D13 post-infection (D13 pi; D105 pv. Significant differences between treatment groups are indicated by asterisks (*p* < 0.05 (*), *p* < 0.01 (**), and *p* < 0.001 (***). One-way analysis of variance (two-sided), Tukey’s pairwise comparisons.

**Figure 10 vaccines-09-00236-f010:**
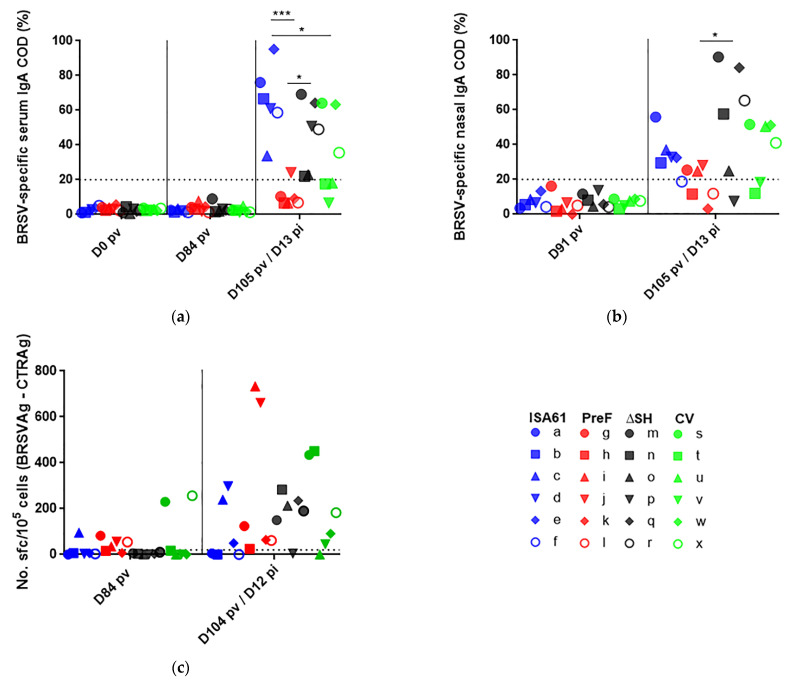
Vaccination with PreF and CV reduced bovine respiratory syncytial virus (BRSV)-specific serum immunoglobulin (Ig)A response to challenge and did not induce detectable interferon (IFN)γ-producing memory T cells in peripheral blood. Calves were vaccinated with either (i) Pre-fusion F (PreF) adjuvanted by ISA 61 VG intramuscularly (i.m.), (ii) ΔSHrBRSV (ΔSH) intranasally (i.n.), or (iii) a commercial vaccine (CV) i.n., or were injected with ISA 61 VG i.m. and were challenged with virulent BRSV, 3 months later. BRSV-specific IgA and IFNγ-producing T cells were determined by enzyme-linked immunosorbent assay (ELISA) and IFNγ- enzyme-linked immune absorbant spot (ELI)spot, respectively, in samples collected before and after BRSV-challenge. Days (D) are indicated as post-vaccination (pv) and post-infection (pi). Corrected optical density (COD) values of (**a**) BRSV-specific serum IgA and (**b**) nasal IgA are presented as percent of positive reference sera with a titer of 1:12,800. BRSV-specific IFNγ-producing memory T cells were analyzed by ELISpot of γ/δ-depleted peripheral blood mononuclear cells, stimulated either with heat-inactivated BRSV-infected cell lysate (BRSVAg) or uninfected cell lysate (CTRAg). (**c**) IFNγ-producing memory T cells expressed as number of spot-forming cells (sfc) per 10^5^ cells, obtained with CTRAg subtracted from those obtained with BRSVAg. Significant differences between treatment groups are indicated by asterisks (*p* < 0.05 (*) and *p* < 0.001 (***). One-way analysis of variance (two-sided), Tukey’s pairwise comparisons. Dotted lines represent the limits of detection.

**Figure 11 vaccines-09-00236-f011:**
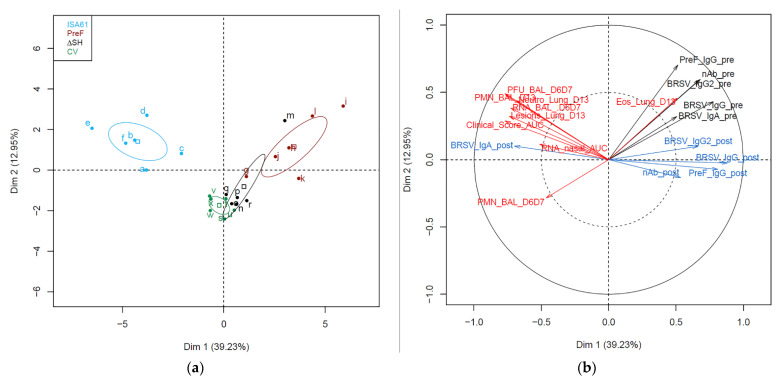
Pattern of immunological and disease parameters. Calves were vaccinated with either (i) Pre-fusion F (PreF) adjuvanted by ISA 61 VG intramuscularly (i.m.), (ii) ΔSHrBRSV (ΔSH) intranasally (i.n.), or (iii) a commercial vaccine (CV) i.n., or were injected with ISA 61 VG i.m. and were challenged with virulent BRSV, 3 months later. (**a**) Principal component analysis plot of the clinical, virological, and immunological data of calves represented as dots and 95% confidence ellipses. Dimension (Dim) 1 explains 39.23% of the total variation of the data between calves, and Dim 2 explained a further 12.95% of the variation. (**b**) Loading plot of the same data as in (**a**). In red the virological, inflammatory, and clinical data; in black the immunological parameters measured at D84 post-vaccination (pv, _pre) and in blue the immunological parameters measured at D105 pv/D13 post-infection (_post).

**Table 1 vaccines-09-00236-t001:** Peak rectal temperatures and respiratory rates of vaccinated calves and controls after the bovine respiratory syncytial virus (BRSV) challenge. Calves were vaccinated with either (i) Pre-fusion F (PreF) adjuvanted by ISA 61 VG intramuscularly (i.m.), (ii) ΔSHrBRSV (ΔSH) intranasally (i.n.), or (iii) a commercial vaccine (CV) i.n., or were injected with ISA 61 VG i.m. and were challenged with virulent BRSV, 3 months later and monitored 13 days post-infection.

Clinical Signs	ISA61	PreF	ΔSH	CV
Rectal temperature mean of peak, °C (SD)	40.0 (±0.7)	39.4 (±0.1)	39.5 (±0.2)	39.3 (±0.2) *
Respiratory rate, mean of peak, breath/min (SD)	60 (±20.8)	38 (±3.2) *	40 (±3.3) *	43 (±3.6)

* significant difference in comparison with controls, 0.01 < *p*-value < 0.05.

**Table 2 vaccines-09-00236-t002:** Polymorphonuclear leukocytes in histological lung sections of calves following bovine respiratory syncytial virus (BRSV) challenge. Calves were vaccinated with either (i) Pre-fusion F (PreF) adjuvanted by ISA 61 VG intramuscularly (i.m.), (ii) ΔSHrBRSV (ΔSH) intranasally (i.n.), or (iii) a commercial vaccine (CV) i.n., or were injected with ISA 61 VG i.m. and were challenged with virulent BRSV, 3 months later. Histopathology was performed on day 13 post-infection.

Group	id	Neutrophil Score ^a^			Eosinophil
		Intra- Epithelial	Lumen Bronchi Bronchiole	Lumen Alveoli	Number ^b^
ISA61	a	(+)	-	(+)	2
	b	++	++	++	11
	c	+	(+)	(+)	40
	d	++	++	+	24
	e	+	++	++	21
	f	-	+	+	10
PreF	g	++	+	-	2
	h	+	-	-	8
	i	-	-	-	114
	j	-	-	-	17
	k	-	-	-	105
	l	-	-	-	174
ΔSH	m	-	(+)	-	3
	n	-	-	-	21
	o	-	-	-	14
	p	-	-	-	13
	q	-	-	-	22
	r	-	-	-	7
CV	s	-	-	-	5
	t	-	-	-	6
	u	n.a.	n.a.	n.a.	n.a.
	v	-	-	-	2
	w	-	-	-	0
	x	+	+	(+)	0

^a^ Grading of neutrophilic infiltrate in lung tissue. - absence, (+) insignificant, + mild, ++ moderate, +++ severe, n.a.; not analyzed ^b^ sum of eosinophils in 20 high power fields (area of 4.74 mm^2^).

**Table 3 vaccines-09-00236-t003:** Top correlations between the proteins in bronchoalveolar lavage (BAL) obtained from PreF-, ΔSHrBRSV-, or CV-immunized calves and controls before bovine respiratory syncytial (BRSV) challenge (day (D)-1/D0 pi), and disease parameters after the challenge in the same calves.

PreF			ΔSHrBRSV			CV		
Id ^a^	Disease Parameter	Corr ^b.c^	Id ^a^	Disease Parameter	Corr ^b.c^	Id ^a^	Disease Parameter	Corr ^b.c^
Ig	Clincal score AUC	−0.85	ALPL	BAL PMN D13	−0.78	ORM1	nasal BRSV RNA AUC	−0.78
A5D7Q2 ^d^	BAL BRSV live D13	−0.82		BAL BRSV live D13	−0.78	GSN	BAL PMN D6/7	−0.75
	BAL BRSV RNA D6/7	−0.8		BAL PMN D6/7	−0.78	PGLYRP1	Eos lung D13	−0.78
	Lung lesions D13	−0.8		BAL BRSV RNA D13	−0.78	SCGB1A1	PMN lung D13	−0.94
	nasal BRSV RNA AUC	−0.78		BAL BRSV RNA D6/7	−0.77		BAL PMN D13	−0.77
ALPL	BAL BRSV RNA D13	−0.82	ARHGDIB	BAL BRSV RNA D13	−0.77			
ANPEP	BAL BRSV live D13	−0.78	CD44	PMN lung D13	−0.75			
G5E5H2 ^e^	nasal BRSV RNA AUC	−0.8	GPI	BAL PMN D6/7	−0.82			
	Clinical score AUC	−0.77		BAL BRSV RNA D6/7	−0.79			
LYS	BAL BRSV live D13	−0.78		Clinical score AUC	−0.76			
	BAL BRSV RNA D6/7	−0.77	IGL	BAL PMN D6/7	−0.77			
AOX2	EOS LUNG D13	−0.76	LDHB	BAL PMN D6/7	−0.75			
S100A12	BAL BRSV RNA D6/7	−0.75	PGLS	BAL PMN D6/7	−0.77			
S100A8	PMN lung D13	−0.83	PIGR	Eos lung D13	−0.75			
SCGB2A2	BAL BRSV RNA D6/7	−0.75	WFDC2	BAL BRSV RNA D13	−0.89			
	nasal BRSV RNA AUC	−0.74						
WFDC2	BAL BRSV RNA D13	−0.85						
	nasal BRSV RNA AUC	−0.77						
	BAL BRSV live D13	−0.76						
	PMN lung D13	−0.75						

^a^ Gene or protein identifier. Data generated by using the threshold ≤−0.75. ^b^ Analyses based on bronchoalveolar lavage (BAL) from all calves for which samples with sufficient quality were available (*n* = 21). ^c^ Spearman’s rank correlation coefficient. ^d^ Immunoglobulin (Ig). Identity obtained through InterPRO (integrated resource of protein families, domains, and functional sites); ^e^ Ig V-set. Identity obtained through InterPRO.

## Data Availability

Proteomic neutrophil-related data presented in this study is openly available in FigShare at https://doi.org/10.6084/m9.figshare.13689214.v1 (accessed on 7 March 2021).

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
