# Peer review of "Single-Shot Vaccines against Bovine Respiratory Syncytial Virus (BRSV): Comparative Evaluation of Long-Term Protection after Immunization in the Presence of BRSV-Specific Maternal Antibodies"

_vaccines, 2021, doi:10.3390/vaccines9030236_

Round 1

Reviewer 1 Report

General comments:

In this paper, Valarcher et al. assessed and compared different vaccine candidates against bovine respiratory syncytial virus (BRSV). Furthermore, vaccine candidates were tested in the presence of maternally derived antibodies (MDA). Pre-fusion protein (PreF) and Montanide ISA61 VG as well as BRSV lacking the SH gene (∆SHrBRSV) showed they are two efficient one-shot candidate vaccines. This is an important demonstration in the field of RSV research. Thus, overall, the article is good and very well-presented. Please find below a detailed list of my minor/moderate remarks/suggestions.

Strengths: Pleasant and well-written. Different suitable technical approaches to assess the vaccines and nice figures. Original research and convincing results

Weakness: Maternal interference could be further discussed for vaccine candidates.

Major

/

Moderate

-L33: Please briefly mention bovine age

-L36: I would replace “ISA61” by “ISA61 VG alone”

-L51: Please mention the family of the virus (Pneumoviridae), this a quite new one. Moreover, please briefly described the viruses in this family (negative RNA…).

-L78-81: Especially as a preclinical model. The paper from Guzman and Taylor could be cited here or later (doi: 10.1016/j.molimm.2014.12.004).

-L105: Please specify the strain, lab or field strain.

-L108: immune parameters. Authors could even write “correlates of protection”.

-L130: Quite lower for ∆SHrBRSV than others.

-L131: The age variation is quite large.

-L154: Please justify the choice of the Snook strain

-L158: Please mention the dose of pentobarbital.

-L197: Please mention if it was Taqman or SYBR Green for the PCR in the current paper.

-2.9: How was tested the normality of the distribution? Please specify. I guess it was using a Shapiro-Wilk test or similar.

-Figure 4A: High proportion of granulocytes in the BAL of CV immunized bovines. This point should be further discussed.

-L507: The variation between groups is really high. Please comment.

-Major reviews about maternal interference must be cited and discuss in the discussion section. Some hypotheses could be made for the current study. See doi: 10.1016/s0264-410x(03)00342-6, doi: 10.3389/fimmu.2014.00446

-Authors could also use the following paper in another species to enrich their discussion, see doi: 10.1016/j.vaccine.2013.05.008

Minor

-L37: Please replace “sacrificed” by “euthanized”

-In figure 1: IFNg should be replaced by IFNÉ£.

-L200: Common abbreviation like MDBK, NSAID (L386), WAP (L660) should be explained too.

-L202 and elsewhere: Please replace ml by mL.

-L214: 1h, 1 h or 1 hour (L249, 388), please harmonize in the paper.

-L226 and elsewhere (L724): Some double spaces to replace by single spaces.

-L313: Please correct Homo sapiens.

-L406: Please remove the dot at the end, this is not a sentence.

-Some presentation issues in the reference section, see journal names L1017, references 37 and 38 (not abbreviated?).

Author Response

Reviewer #1: Moderate

-L33: Please briefly mention bovine age

- L136-137: the age has been added “At vaccination (when the mean age of the calves was 36 days, range 24-56 days), one group of calves”

-L36: I would replace “ISA61” by “ISA61 VG alone”.

-L36, L137: We have added “alone” and specified that ISA 61 stand for ISA61 VG in text and in the figure legends. We would prefer to keep ISA61 because the submitted work is a follow up of a recent publication (Riffault et al Vaccines 2020, 8, 231; doi:10.3390/vaccines8020231) in which ISA61 was used. By changing the name, the reader might think that we changed the adjuvant. Furthermore, a longer name will make the reading and the layout of the figures and tables more difficult.

-L51: Please mention the family of the virus (Pneumoviridae), this a quite new one. Moreover, please briefly described the viruses in this family (negative RNA…).

-L55: the following sentence and reference were added: “BRSV is an enveloped, non-segmented, negative-stranded RNA virus that belongs to the Orthopneumovirus genus within the Pneumoviridae family.” (reference: ICTV online. https://talk.ictvonline.org/ictv-reports/ictv_online_report/negative-sense-rna-viruses/w/pneumoviridae/738/genus-orthopneumovirus (accessed on 24/02/2021)).

-L78-81: Especially as a preclinical model. The paper from Guzman and Taylor could be cited here or later (doi: 10.1016/j.molimm.2014.12.004).

-L85 The requested reference was added.

-L105: Please specify the strain, lab or field strain.

 -L109: The following information was added: “Calves were challenged as described previously [16] with the Snook strain of BRSV three months after vaccination”.

-L108: immune parameters. Authors could even write “correlates of protection”

-L112: immune parameters has been replaced by “correlates of protection”

-L130: Quite lower for ∆SHrBRSV than others.

-We disagree with the reviewer’s comment. The neutralising MDA titers were not lower in the ∆SHrBRSV-vaccinated calves at vaccination (Figure 8). The text was changed to clarify that the MDA was measured at arrival to the PFIE facilities and that the level of BRSV-neutralizing MDA did not differ at the time of vaccination:

- L134: “the level of BRSV-specific serum MDA on arrival at the PFIE facilities”

-L618: “The BRSV-neutralizing antibody titres did not differ significantly between calves on the day of vaccination. Despite the presence MDA,”

-L131: The age variation is quite large.

To avoid cross contamination of calves with different pathogens, we chose to buy animals from the same herd. It was not possible to get calves with more similar age.

-L154: Please justify the choice of the Snook strain.

The following sentence was added:

-L164: “Based on the high level of clinical signs of disease induced by the Snook strain of BRSV in previous studies, this virus strain was used to test the vaccines (Blodörn et al. 2015)”

-L158: Please mention the dose of pentobarbital.

This has been modified:

-L169: “140 mg/kg intravenously”

-L197: Please mention if it was Taqman or SYBR Green for the PCR in the current paper.

-L209: “Taqman” has been added

-2.9: How was tested the normality of the distribution? Please specify. I guess it was using a Shapiro-Wilk test or similar.

-The normality was tested using the Anderson-Darling test. This has been specified:

 -L348: “The distribution was tested by using Anderson-Darlings test” was added.

-Figure 4A: High proportion of granulocytes in the BAL of CV immunized bovines. This point should be further discussed.

-We agree and we have modified the text as follows:  

-L770: “probably as a result of a better virological protection” was added

-L928:  “The strong neutrophilic response detected D6/7 pi in calves that had been vaccinated with CV may be due to lack of protection against virus replication and/or reduced neutrophil autoregualtion“ was added.

-L507: The variation between groups is really high. Please comment.

-We agree that the virological data varied between groups. This is probably due to the vaccine composition or the route of immunisation. In case the reviewer means within groups, we have amended the text:

-L791: “Some of the variation that was observed among calves immunised with the same vaccine might be related to the level of BRSV-neutralising MDA at immunisation. Neverthelesss, …“.

-Major reviews about maternal interference must be cited and discuss in the discussion section. Some hypotheses could be made for the current study. See doi: 10.1016/s0264-410x(03)00342-6, doi: 10.3389/fimmu.2014.00446

-Authors could also use the following paper in another species to enrich their discussion, see doi: 10.1016/j.vaccine.2013.05.008

-Two of the suggested references have been added and the following change made:

- L840: “Several approaches to overcome the negative effect of MDA on vaccination have been presented [42] and some have been applied in cattle in the past, “ has been modified.

-L876: i) the effect of the adjuvant, as previously observed [14,17,47]; ii) differences in the anti-body repertoire between the mother, which had been previously vaccinated by the parenteral route with a live BRSV vaccine, and calves, as hypothesized previously [12,42] and described in humans [48]; iii) the BRSV MDA were mainly directed against antigenic sites on post F and very little against those present only on PreF (Sites 0 and V), or the small size of the antigen, which prevented inhibition of B cell activation [42].

Minor

-L37: Please replace “sacrificed” by “euthanized”

-L37: “sacrificed” has been replaced by “euthanized”

-In figure 1: IFNg should be replaced by IFNÉ£.

-Figure 1: IFNg was replaced with IFNÉ£.

-L200: Common abbreviation like MDBK, NSAID (L386), WAP (L660) should be explained too.

-L212/723 The meanings of the abbreviations MDBK (Madin-Darby Bovine Kidney) and for WAP (Whey-Acidic Protein) have been specified

-L420: “NSAID” was replaced with “non-steroidal anti-inflammatory drug”

-L202 and elsewhere: Please replace ml by mL.

-“ml” has been replaced by mL throughout the text and in the figures

-L214: 1h, 1 h or 1 hour (L249, 388), please harmonize in the paper.

-1 h is now used throughout the paper.

-L226 and elsewhere (L724): Some double spaces to replace by single spaces.

-This has been amended

-L313: Please correct Homo sapiens.

-L 341: This has been amended

-L406: Please remove the dot at the end, this is not a sentence.

-The dot has been removed.

-Some presentation issues in the reference section, see journal names L1017, references 37 and 38 (not abbreviated?).

- The journals are now abbreviated

Reviewer 2 Report

I sincerely hope my comments are taken as a constructive criticism

All of my comments are included in the attached file

Author Response

Reviewer #2:

L53: When in combination with other pathogens, particularly secondary bacterial infection' Although authors cite few reference, only one mentions that sole-infection may cause significant losses

-The reference was changed.

-BRSV can cause severe disease also without secondary bacterial infection (e.g. doi: 10.1371/journal.pone.0186594). In Sweden we observe regular epizootic BRSV outbreaks with severe disease in adult cattle.

L216: Why underlined?

-The underlining was removed

L319: state when means were used and when medians:

-If means or medians were compared depended on the null hypotheses of the different tests (means for ANOVA and medians for Kruskal-Wallis). We have removed this statement in the text not to confuse the reader to think that medians were shown in the figures. We have left the statement “Data shown are either individual or means with standard deviation (SD)”

-L346. The phrase was changed “Statistical analyses were performed in Minitab version 16, using One-way ANOVA followed by Tukey’s test for normally distributed data, and Kruskal-Wallis test followed by Mann-Whitney test for data which followed a non-normal distribution”

L321: What were the factors taken into account when carrying out the ANOVA - only group?  The repeated effect of an individual calf must be accounted for.  e.g., It is likely that calf that had a high response early in the post-vaccination period was still high at the end of the study period.

-As the main interest in this study was to compare different groups, and the difference both for groups and for individuals between the time points were quite large, we decided to do the analysis individually for each day instead of setting up a repeated measures ANOVA. I.e. for each analysis the fixed effect in the model is only group and we made a separate analysis per time point.

L325What was the level of significance.  I assume P<0.05.  However, it would be good to be specified.

-L350: The following sentence was added “The cut off for significance was set to <0.05.”

L359: 'statistical' or 'statistic' are not essential throughout the text. It is different or not.:

-L390: “difference” has been added.

-Statistical and statistic have been removed throughout the text and figure legends.

L362: 'Kruskal-Wallis':

-L396 “-“ has been added.

L380: It would be good to stipulate if it was inspiratory or expiratory dyspnea (i.e. differentiation between upper and lower respiratory tract):

-L419 “and expiratory” has been added

L380: 'increased' / 'enhanced':

-L414: “enforced” has been replaced by “enhanced”

L381: better use obtundancy (a sign) rather than depression (a symptom - symptom by definition is a subjective feeling as expressed by the patient.  As calves do not talk, a sign is better than symptom.

-Even if depressed is used in many articles and veterinary text books (since in animals it rather translates a demeanour than the depression state like in human), the authors preferred to replace “depressed” by “decreased demeanor” (L416), which corresponds more to reality.

L384: 'rate':

-L418: “s” has been removed

L395: 'rate':

-L436: “s” has been removed

L397: vaccines. The highest ...':

-L438: the text was split in two sentences as requested

L516: 'were' (data is plural. A 'datum' is singular):

-L564: this has been corrected

L716-726: Would be good to stipulate here that in future virologic data may be used and non need of euthanasia may be required (this is in-line with Animal Welfare).  Thus further experiments may be carried out in 'commercial settings'.

Monitoring only virus shedding in a field setting would be too costly and would additionally need to be combined with clinical scoring, necropsy of dead animals, serology of sentinels, etc. Therefore, we disagree that this is appropriate to suggest. We think it would be better to find correlates of protections before challenge to avoid euthanasia, as suggested by the reviewer.

L728: 'Mild':

-L796 “slight” was replaced by “mild”

L756: Authors may want to challenge the notion not to vaccinate calves with maternal immunity against other disorders as well.  However, this is not essential.

-No change was made

 L789: Isn't this indication that intranasal vaccines stimulated cell-mediated local immunity with sensitised lymphocytes that reacted rapidly on the next contact with the antigen?

-We agree with the reviewer and have modified the text as follows:

-L861: “vaccine-induced” anamnestic response was added

L853: Isn't worth mentioning that the 3 months may be very important for the feedlot industries.  If calves were to be vaccinated at the backgrounding operation 2-4 weeks before transport, they would be protected for the duration of the most critical period (including transport and first 4-6 weeks at the feedlot - the period when the BRDC is with a highest incidence).

-The following text was added:

-L933: “If administered before grouping of animals, this covers the most critical period in feedlot production.“
